# Elastic Cytomatrix Dynamics Influences Metabolic Rate and Tumor Microenvironment Formation

**DOI:** 10.3390/cancers17223686

**Published:** 2025-11-18

**Authors:** Tattym E. Shaiken, Tulendy T. Nurkenov, Meruyert S. Kurmanbayeva, David Y. Graham

**Affiliations:** 1Department of Medicine, Baylor College of Medicine, One Baylor Plaza, Houston, TX 77030, USA; dgraham@bcm.edu; 2PeriNuc Labs, University of Houston, Technology Bridge, Houston, TX 77023, USA; 3Center for Cytomatrix Research, Al-Farabi Kazakh National University, Almaty 050040, Kazakhstan; 4Department of Internal Medicine, University of Texas Health Science Center, Houston, TX 77030, USA; 5Section of Gastroenterology, Michael E. DeBakey VA Medical Center, Houston, TX 77030, USA

**Keywords:** cytomatrix, Warburg effect, actin, tumor microenvironment, metabolism, extracellular matrix, malignant transformation

## Abstract

Traditionally, the chemical reactions that occur within cells are viewed as processes in an aqueous solution with free diffusion. However, the cytoplasm is organized as a two-phase system consisting of a viscous fluid, the cytosol, and an elastic solid matrix, the cytomatrix. The solid phase contains immobilized enzymes that operate without spatial constraints, enabling chemical processes to occur simultaneously without interference. The dynamics of the cytomatrix regulate the rate of metabolism through cytosolic movement, manifesting as cytoplasmic fluctuations that are more intense in cancer cells. The intracellular cytomatrix interacts with the extracellular matrix (ECM). The ECM proteoglycans and glycoproteins produced by the cytomatrix serve as markers for tissue cells and are crucial for cell recognition, signaling, and immune surveillance and response. Translocation of mutated ECM proteins to the cell surface may trigger an immune response, attract various stromal cells, and form a tumor microenvironment.

## 1. Introduction

Every living organism, from bacteria to humans, has a distinct, recognizable physical shape, and this also applies to every cell in a tissue at the microscopic level, which performs a specific function because of its physical attributes. Disruption of the cell’s physical morphology, known as dedifferentiation, is often observed in cancers. The role of the solid phase in maintaining cell morphology as a structural and functional anchor within a large liquid environment remains to be elucidated. It remains uncertain whether cells need a solid phase—defined as a solid component that serves as a stationary or solid-bound phase distinct from the liquid phase—to maintain cell shape and size. The prevailing view is that the cytoskeleton, which consists of cytoplasmic filaments including microfilaments, intermediate filaments, and microtubules, provides structural support and helps maintain cell shape. However, the cytoskeleton fibers are highly dynamic, such that the cytoskeleton is more of a metaphor than a rigid construction, like the skeletal system of the human body. It is also not clear how a solid structure based on cytoplasmic fibers that form a highly complex cytoarchitecture could be isolated and characterized as an elastic yet solid framework. Supramolecular complexes or aggregates typically form a more solid structure with dynamic filaments [1]. This construction, known as mechanical integration through cooperative architecture, involves the integrated structure that coordinates cell responses via shared structural and signaling frameworks and acts as a unified system [2,3]. We previously demonstrated that the integrated cooperative cytoarchitecture, named the cytomatrix, which forms an operating system for the cell, where structure and function are inseparable, can be chemically isolated [4]. In this review, we examined the biochemical composition of the cytomatrix and its connection to the extracellular matrix (ECM), focusing on the unique features of chemical processes within the cytoplasm and the role of microfilaments in regulating metabolic rates. We also analyzed the energy needs of both healthy and malignant cells in relation to the dynamics of the cytomatrix, explaining the Warburg effect as an adaptation to the rapid fluctuations in energy observed in cancer cells due to solid-phase mechanics. Summarizing known traits of cancer cells and our findings regarding gaps in cell structure (as a two-phase system), along with the role of the Warburg effect in non-muscle actomyosin dynamics, we introduced the Cancer Cycle concept. This concept integrates genetic alterations linked to cancer with the physicochemical processes occurring in the cytoplasm. The intracellular cytomatrix and extracellular matrix, functioning as an integrated solid-phase system, regulate the metabolic rates of both healthy and malignant cells. Changes in the ECM of malignant cells, driven by mutations in the cytomatrix, may attract immune and stromal cells, contributing to tumor microenvironment formation. Finally, we explored the potential of artificial intelligence and neural machine learning in interpreting cells as structured solids containing liquids, which could lead to a new interdisciplinary field combining life sciences with AI technologies.

### 1.1. The Solid Component of the Cytoplasm—The Cytomatrix

Cell size and shape are related to the solid and liquid phases of the cytoplasm. The solid phase, the cytomatrix, is built on three types of cytoplasmic fibers: microtubules, intermediate filaments, and microfilaments and is responsible for maintaining the consistent structural composition of different tissue types and across species [4]. The cytomatrix binds and localizes a diverse spectrum of enzymes, signaling and cancer pathways, protein components of organelles, and extracellular matrix proteins (Figure 1).

All three types of cytoplasmic fibers interact with one another and with various protein complexes, including ribosomes, catalytic complexes, and ribonucleoprotein complexes, and overall are responsible for the shape and size of differentiated cells that form the cytomatrix. Thus, the cytomatrix is structurally complex and serves as a platform assembled from the proteinaceous remnants of organelles, the plasma membrane, the inner nuclear membrane (in part), and the extracellular matrix (ECM).

The fact that the catalytic complexes are immobilized in the cytomatrix and separated from one another means that the innumerable diverse chemical reactions can only occur simultaneously when the liquid phase, the cytosol, is in motion. The cytosol in motion brings substrates and regulators to catalytic complexes, similar to blood delivering nutrients to organs and tissues, the same biological principle, but on a molecular level. Non-muscle motor proteins actin and myosin serve as a pump, like the heart, which pumps blood throughout the body [5].

### 1.2. Cell Cytoplasm Complexity and Chemical Reactions

Classical biochemistry is founded on assumptions that are valid only in dilute aqueous solutions. These assumptions, when extended to the interior milieu of cells, allow a comprehensive understanding of chemical processes within living cells, particularly in relation to their metabolic functions [6]. Cells are typically tightly packed with molecules [7]. Mammalian cells enclose a highly viscous, crowded, and heterogeneous gel-like environment that limits free diffusion. Despite the crowded environment, individual groups of molecules are not abundant, and no single molecular species is necessarily always present at high concentration [8].

The spatial obstacles and low concentration of individual groups of molecules challenge the applicability of the law of mass action, one of the fundamental principles of chemistry. Although classical biochemists have acknowledged this paradox, they have been unable to provide alternatives other than to describe the chemical reactions observed as if they were occurring in a buffered aqueous solution, and adhered to the laws of thermodynamics.

Despite the unique characteristics of the cellular environment, chemical processes within the cell must be governed by the principles of chemistry and physics, with catalytic processes being regulated by the viscosity, molecular diversity, and spatial constraints. The turnover of catalytic processes is high. The primary regulators of catalytic reactions include the physical and chemical properties of the cytoplasm and feedback regulation. The rate of chemical reactions (i.e., the metabolism) is highly tuned.

Mammalian cells do not experience deficiencies in nutrients; thus, cellular chemical processes depend on a host of external factors, including hormones, growth factors, and signaling nutrients, to determine the cell’s phases of resting, growth, and proliferation. These signals also fine-tune the cell’s metabolism to overcome viscosity and spatial hindrances.

Cell functions can be considered like an orchestra where many different musical instruments work together to produce unique sounds. Like instruments, cell surface receptors respond to specific ligands and hormones in a manner similar to the way the conductor listens to and manages the orchestra. When the communication chain is disrupted (i.e., the conductor fails to maintain harmony), a homeostatic imbalance occurs, leading to the breakdown of the cellular “score”. Which intracellular molecules serve as conductors is a central question.

Tubulin and actin are two motor proteins that “listen to and conduct” cell signaling, thus reflecting the active force of the cell. Both tubulin and actin have enzymatic activity and utilize nucleotides as substrates. Tubulin relies on GTP to form microtubules and direct the movement of motor proteins, including kinesin and dynein. In contrast, the actin superfamily of proteins converts chemical energy from ATP into mechanical energy. Differences in nucleotide preference (ATP vs. GTP) and in filament dynamics indicate that actin is the primary force driving cytoplasmic fluctuations and in regulating the rate of chemical processes in the cell.

### 1.3. Actin Regulation Pathways

Actin is one of the most conserved proteins throughout the evolution of eukaryotes, and one of the most abundant proteins in eukaryotic cells. Actin filaments form a network throughout the cell and are particularly densely concentrated near the plasma membrane (Figure 2) [9]. Actin filaments also arrange themselves in concentric patterns around the nucleus [10].

Actin’s specific localizations are related to its function of contraction and relaxation. Actin dynamics can alter the viscosity of the cytoplasm and the diffusion limitations present in the crowded cellular environment [11]. The complex system functions using the cortical actin microfilaments underlying the plasma membrane, where the polymerization of actin filaments provides the force for a variety of cellular processes through the Ras family of GTPases.

The Ras family proteins localize to the inner leaflet of the plasma membrane, which is crucial for their role in cell signaling pathways [12]. The Ras and Rho family small GTPases induce PI3K activity [12]. The p85a regulatory subunit of PI3K interacts with the small GTPase Cdc42, contributing to both PI3K activation and Cdc42 activation. The latter activates N-WASP, stimulating actin polymerization by interacting with the Arp2/3 complex [13] (Figure 3).

The Arp2/3 complex is an actin filament nucleation and branching machinery that generates pushing forces, driving cellular processes ranging from membrane remodeling to cell and organelle motility. Several upstream signals regulate these processes by directly inhibiting or activating the Arp2/3 complex and by stabilizing or disassembling branched networks [15]. The complex network of actin regulators and nucleation-promoting factors (NPFs) regulates the speed of chemical reactions in the viscous environments of the cytoplasm and nucleus, rather than in movement within organs and tissues, which does not occur.

### 1.4. Role of Microfilaments in Cytomatrix Formation

Actin’s role in the fine regulation of metabolic processes is thought to be related to the mechanics of the solid phase to which it belongs. Actin is one of the most regulated proteins in eukaryotic cells and interacts with various actin-binding proteins (ABPs). ABPs are divided into seven groups according to their specific functions on actin polymers and/or monomers: (1) actin nucleators that promote actin nucleation; (2) regulators of G-actin polymerization/F-actin depolymerization; (3) capping and actin severing proteins; (4) bundling and cross-linking proteins; (5) motor proteins that generate force like myosin; (6) anchoring actin filaments to the plasma membrane; (7) F-actin stabilizing and regulatory proteins [16].

The function of actin is also regulated by post-translational modifications (PTMs) of which more than 140 PTMs have been described in actin sequences of 94 different amino acid side chains [17,18]. There are 19 types of PTMs at numerous sites. Actin is also covalently modified by sugars, the best-known of which are N-linked and O-linked glycosylation, whose sheer number poses a serious challenge to the comprehensive understanding of their role in regulatory mechanisms. Actin generates dynamic filaments composed of geometrically conserved building blocks, often repeated over several thousand times, that give rise to structures decorated with potentially structure-disturbing post-translational modifications. In addition, G- and F-actin interact in a dynamic and rigorously controlled manner with a plethora of ABPs while carrying a large number of modifications [18]. Together, these post-translational modifications and ABPs in the actin filaments create an integrated and dynamic architecture of the cytomatrix.

Cellular processes require tight coordination between microtubules and actin filaments [19]. Microtubules couple with actin microfilaments through CLIP-170, which binds tightly to formins to accelerate actin filament elongation. Crosstalk between microtubules and actin through spectraplakins (proteins that crosslink microtubules and actin), as well as the growth-arrest-specific 2 (GAS2) family proteins that resemble spectraplakins, is essential for cell polarization and dynamics [20,21]. Any overview of the physical mechanisms through which actin and microtubules regulate one another’s dynamics and organization involves multifaceted actions, including crosslinking, guidance of microtubule growth, actin-mediated anchoring, stabilization of microtubule ends, as well as the prevention of microtubule targeting to the plasma membrane as a physical barrier (Figure 2). The process also includes microtubule-mediated nucleation of actin filaments, cell polarity, and mechanical cooperation [22].

Intermediate filaments resist tensile and compressive forces in cells and crosslink with each other as well as with actin filaments and microtubules using desmin, filamin C, plectin, and lamins to form intrinsic nucleo-cytoplasmic networks [23]. Keratin filament turnover is a multistep process that is involved in solid phase motion and facilitates rapid network remodeling in relation to actin and microtubule networks, particularly in epithelial cells [24].

All types of cellular filaments, microtubules, intermediate filaments, and microfilaments, interact with one another to create networks to provide the solid foundation that physically and chemically connects cellular components. This solid phase links internal protein complexes and organelles to the external environment (extracellular matrix) and forms the cytomatrix.

In the cytomatrix, actin functions as a motor protein regulating the fine-tuning of chemical reactions via a plethora of regulators and by organizing the solid phase network [4]. Actin’s role in regulating the rate of chemical reactions in the solid phase cytomatrix is primarily associated with its peripheral localization under the plasma membrane and around the nucleus (Figure 4).

It appears plausible that actin and myosin motility can transmit external signals to internal actin filaments in order to coordinate with Transmembrane Actin-Associated Nuclear (TAN) Lines and Perinuclear Actin Caps and regulate nuclear reactions in connection with nuclear actin and nuclear matrix [10].

### 1.5. Life Cycle of the Healthy Cell

Gene expression, which is governed by external signals, determines the shape, size, and function of cells through chemical processes occurring in the cytoplasm. The cytomatrix, the solid phase of cytoplasm, is dynamic and requires an energy input to maintain the cell’s specific shape and function. In healthy cells, mitochondrial oxidative phosphorylation supplies the energy required to support cytomatrix mechanics, which regulate steady-state synthetic processes and metabolism (Figure 5A). External factors determine the cell’s fate. Cells may enter the resting phase (G0) or proceed to the cell cycle. Entering the cell cycle precedes the accumulation of biomass.

Individual cells may remain in a resting state or enter the cell cycle. External signals such as growth factors or mechanical cues activate cyclins and CDKs, triggering the cell to enter the G1 phase. After passing the G1 restriction point, the cell significantly increases its use of biosynthetic pathways to enhance ribosome production, nutrient intake, and mitochondrial activity, thereby increasing support for S-phase (synthetic phase) requirements and ultimately facilitating mitosis.

In rapidly growing malignant cells, additional energy is required, which is supplied by glycolysis through a phenomenon known as the Warburg effect, thereby energizing enhanced cytomatrix mechanics and biosynthetic processes.

### 1.6. The Warburg Effect as an Adaptation to Rapid Fluctuations in Energy Demand

Malignant tumor cells utilize up to 30 times more glucose than healthy cells and thus produce up to 43 times more lactic acid, irrespective of the oxygen supply [25]. Otto Warburg theorized that the dependence of cancer cells on less productive glycolysis might be due to the malfunctioning of mitochondria. However, further research revealed that the mitochondria were functioning normally and the process was redefined as aerobic glycolysis [26].

An international team of scientists reported the effects of disabling Warburg effect-associated lactic acid production without altering the rate of cell division [27]. Cells consumed more oxygen, suggesting that the mitochondria were producing more ATP. They concluded that the Warburg effect is not essential for obtaining biomass precursors and highlighted its role in energy generation (e.g., oxygen consumption increases in Warburg-null cells) [27]. However, the exact destination of the excess energy produced remained uncertain.

Mitochondrial oxidative phosphorylation and glycolysis are two dynamic sources of ATP production, exquisitely regulated systems that adapt to cellular demands [28]. Oxidative phosphorylation occurs in the mitochondria and generates a significant amount of ATP by utilizing energy from the electron transport chain and the proton gradient. In contrast, glycolysis takes place in the cytosol, where glucose is broken down into pyruvate. This process produces a smaller amount of ATP along with two molecules of NADH per glucose molecule. Glycolysis does not require oxygen, making it essential for energy production under anaerobic conditions. Pyruvate can then be directed into the mitochondria for oxidation or converted into lactate (Figure 5B).

Oxidative phosphorylation happens across the inner mitochondrial membrane and is driven by the electron transport chain (ETC), which uses NADH and FADH_2_ as electron donors. This process creates a proton gradient across the membrane, which powers ATP synthase [29]. Oxygen serves as the terminal electron acceptor in this process. The NADH produced during glycolysis (and later in the citric acid cycle) feeds into the ETC, facilitating a smooth metabolic handoff between these pathways.

In tumors, the Warburg effect provides a way to adapt to the rapid changes in energy requirements of cancer cells [30]. Muscle cells are the prime example of the need to respond to rapid fluctuations in energy demand. Muscle cells utilize the ability of the glycolytic pathway to upregulate ATP production 100-fold. However, tumor cells do not appear to participate in either physical activity or mechanical work and the consequences of the energy generated through aerobic “tumor” glycolysis have not been identified experimentally. It remains unclear where the surplus energy generated by the Warburg effect is utilized by tumor cells. One possibility is that, similar to muscle cells, malignant cells use this energy for mechanical activity and produce lactate as a byproduct.

### 1.7. Actin Microfilaments Are a Consumer of the Energy Surplus

Our recent studies suggest that non-muscle actin and myosin consume the energy surplus generated by aerobic glycolysis in cancer cells [5]. Malignant cells may require more energy than can be produced by the limited number of mitochondria through aerobic glycolysis. We hypothesized that the ATP generated by glycolysis could fulfill the energy requirements for the micromechanics of the cytomatrix, specifically for actin. To test the hypothesis, actin and myosin modulators were examined for their impact on lactate production in the HCT-15 colon cancer cell lines (Figure 6).

Wiskostatin, Blebbistatin, and Narciclasine exhibited varying degrees of suppression in lactic acid production (Figure 6A,C). Wiskostatin selectively suppresses WASP-mediated actin polymerization [5]. Narciclasine impairs the organization of the actin cytoskeleton by targeting RhoA GTPases [31]. Blebbistatin, an ATPase inhibitor selective for myosin II [32]. On the contrary, SMIFH2 and CK 666, actin modulators, increased lactate generation in colon cancer cells (Figure 6D,E). SMIFH2 inhibits actin polymerization by binding formins [33] and suppresses myosin ATPase activity [34].

CK 666 disrupts actin polymerization by locking the Arp2/3 complex in an inactive conformation [35] and protects against ferroptosis, the iron-dependent form of programmed cell death [36]. Myosin Light Chain Kinase (MLCK) Inhibitor, ML-7, also increases L-lactate generation (Figure 6F). MLCK regulates contractions through myosin activation [37]. ML-7 has been shown to protect cardiac function by increasing energy production [38].

Thus, modulators of actomyosin activity can both increase and decrease lactate production. Compounds that increase lactic acid generation are involved in modulating several additional pathways, which have a protective role by increasing energy production.

We linked the non-muscular cells’ actomyosin with aerobic glycolysis. Intracellular chemical processes are influenced by cytoplasmic fluctuations driven by the dynamics of the cytomatrix. In this context, the cytoplasm’s viscosity is an essential factor in regulating cellular chemical processes in coordination with motor proteins to achieve a physiological rate of metabolism. The actin superfamily, which includes conventional actin and actin-related proteins (ARPs), harnesses ATP hydrolysis to drive dynamic structural changes that produce mechanical force.

### 1.8. Non-Muscle Actin Dynamics and Cytoplasmic Fluctuations

Actin microfilament dynamics occur rapidly in cells. Actin filaments polymerize and depolymerize at rates of up to several hundred subunits per second. Both reactions require ATP (adenosine triphosphate) hydrolysis. The proteins linked to filaments can rearrange within milliseconds, and function as molecular motors, being able to undergo cyclical conformational changes at rates exceeding 100 Hz [39].

Non-muscle actin dynamics in live cells is an area that has been intensively studied over the last decade using novel imaging technologies. Fluctuations or movements within the cytoplasm, previously thought to be random thermal motion, appear to be directed. Force spectrum live cell microscopy (FSM) demonstrated that fluctuations in the cytoplasm arise from active forces and are significantly greater in malignant than in benign cells [40]. Blebbistatin, the myosin inhibitor, markedly decreased the mean-square displacement (MSD) of injected colloidal particles. The depletion of ATP showed that the motion of labeled particles is an ATP-dependent process. Live-cell lattice light-sheet microscopy (LLSM) time-lapse images have recently demonstrated vigorous actin dynamics in Caco2 cells expressing LifeAct-RFP, which are represented in 3D [41]. Molecular motors in cells typically produce highly directed motion, with the sum of this motor protein activity being visualized as random fluctuations of the cytoplasm.

### 1.9. The Role of the Cytomatrix in Malignant Transformation

Using the KEGG database, the solid-phase cytomatrix and the liquid-phase cytosol were analyzed (Figure 7). Both cytosol and cytomatrix contain ribosomes. The Ribo-seq showed differences in protein translation between the cytomatrix and cytosol. All these pathways, as shown in Figure 7A,B, play significant roles in the regulation of actin microfilament activity, either directly or indirectly. Cell adhesion, tight junctions, gap junctions, focal adhesions, and adherence junctions are all intimately linked with the actin microfilaments (Figure 4B). Additionally, endocytosis and regulatory pathways involving actin are overrepresented in the cytomatrix and underrepresented in the cytosol.

The cytomatrix is especially enriched with signaling pathways associated with cancer (Figure 7A) and was first identified in retroviruses [42]. For example, the HER/ErbB family of growth factor receptors contains an extracellular region that binds a ligand, a transmembrane domain, and an intracellular catalytic domain. Ligand binding induces a conformational change in the extracellular domain, promoting HER/ErbB protein homo- or heterodimerization and transphosphorylation of intracellular domains [43]. ErbB receptors, particularly ErbB2, are involved in regulating actin microfilament dynamics [44]. The signaling pathways associated with ErbB, such as the MAPK and PI3K-AKT pathways, play critical roles in actin remodeling, which is essential for changes in cell shape [45,46,47]. ErbB-Src signaling is a critical interaction where ErbB receptor activation leads to the recruitment and activation of Src family kinases. Activating mutations in c-Src are accompanied by dramatic changes in the structure of actin microfilaments. Cortactin is a substrate for Src. The adaptor Nck is an important component of this system that links phosphorylated cortactin with N-WASp and WASp-interacting protein (WIP) to activate the Arp2/3 complex, which results in actin assembly [48]. The cytosol is enriched with the pathways of nucleotide, amino acid, and protein degradation as well as carbohydrate metabolic pathways (Figure 7C,D).

### 1.10. Relationship Between Cell Growth, Size, and Cell Division

Cell size is fundamental to cell physiology because it sets the scale of intracellular geometry, organelles, and biosynthetic processes. In animal cells, size homeostasis is controlled through two phenomenologically distinct mechanisms [49]. First, size-dependent cell cycle progression ensures that smaller cells delay cell cycle progression to accumulate more biomass than larger cells prior to cell division. Second, size-dependent cell growth ensures that larger and smaller cells grow slower per unit mass than more optimally sized cells. Thus, homeostatic cell size control is achieved via the coupling of cell size, cell growth rate, and cell cycle progression.

While sizes vary among myocytes, neurons, and adipocytes, each is suited to its function (e.g., muscle cells are smaller in the face than in the legs) [50]. Increased cell size is not only a consequence but also a cause of permanent cell cycle exit. Senescent cells tend to be larger because they have accumulated biomass without dividing [51].

Size and growth rate are regulated by the balance of macromolecule accumulation and loss. Despite rapid turnover of components, healthy cell size maintenance can persist throughout a lifetime, and uniformity of cell size is a consistent feature of healthy tissues [52] (Figure 8A). Cell size is defined by a cell’s total macromolecular protein mass, as this metric most closely reflects the sum of anabolic processes typically associated with cell growth and with activity in growth-promoting pathways such as mTORC1.

Growth rate and cell cycle length are coordinated to maintain cell size at fixed values. A confirmation of this dual-mechanism model is that perturbations that lengthen the cell cycle would be counteracted by a compensatory decrease in growth rate, allowing cells to accumulate the same amount of mass despite the longer periods of growth. Conversely, perturbations that reduce growth rate would be counteracted by a compensatory lengthening of G1 [52].

Cell density shows very little variation within a given cell type. The dense packing of macromolecules inside the cytoplasm creates a number of emergent properties summarized under the concept of molecular crowding [53]. Macromolecular crowding has a profound impact, positive or negative, depending on context. For example, local crowding increases, but general crowding of molecules without gathering specific molecules at a specific location decreases reaction rates. Healthy cells usually regulate their size (mass), but in some cancers, this control is disrupted, leading to increased size variability. For instance, in lung adenocarcinomas, this loss of size control may be significant [54]. In preneoplastic esophageal and breast cancer cells, larger volumes are noted in more advanced stages, indicating a link between cell size and cancer progression.

Thus, cell size, biosynthesis, and cell division are intricately linked processes, particularly in the context of cancer, where cancer cells exhibit altered regulation of these aspects. Cell size is a result of growth (increase in mass or volume) and division (splitting into two). The elastic nature of the solid-phase cytomatrix allows for a certain degree of cell expansion, exceeding which may promote cell division in part by the influence of surface-volume ratios. Surface to volume ratio determines metabolic activity of a cell (i.e., 4 pr^2^ for surface vs. 4/3 pr^3^ for volume). Small cells have more surface area relative to their volume, which means efficient exchange of nutrients, gases, and waste. This leads to higher metabolic rates, since materials for metabolism are quickly absorbed. As cell size increases, volume grows faster than surface area, limiting exchange with the environment. This results in metabolic constraints, forcing cells to either slow their metabolic activity or evolve adaptations (e.g., invaginations or specialized transport) or enter the senescent state. This cellular response, which is characterized by an irreversible cell cycle arrest after limited cell divisions, is called replicative senescence, also known as “the Hayflick limit.” [55,56].

### 1.11. Concept of Cancer Cycle

Genetic and epigenetic abnormalities significantly impact cellular homeostasis (i.e., the process by which a cell maintains a stable internal environment in response to changing conditions). Mutations, epigenetic changes, and some pathogens alter homeostasis and influence actin filament assembly or disassembly, either directly or indirectly, leading to a less differentiated state. Actin microfilament dynamics stimulate cytomatrix mechanics (actin is part of the cytomatrix), which manifests as cytoplasmic fluctuations. Subsequently, increased cytosol movement through the cytomatrix speeds up chemical reactions in the cell by enhancing substrate delivery to catalytic complexes. Due to the rapid acceleration of biosynthetic processes, excessive biomass accumulation from the overproduction of proteins and other substances leads to overcrowding within the cell and an increase in cell size (mass) [57]. Size uniformity of animal cells is regulated by the cell cycle phase G1-length [58]. During G1, the cell increases in size (mass), synthesizes mRNA and proteins, and duplicates organelles in preparation for DNA replication. To maintain cell size and homeostasis, cell survival hinges on its ability to divide and continue its life cycle or enter the senescent state [52].

Thus, tumor cell division is a consequence of alterations in biosynthesis and cell size (mass), in which actin dynamics require energy, which is supplied by aerobic glycolysis, producing lactate as a byproduct. The development of tumors from mutated cells requires at least four conditions, beginning with genetic and/or epigenetic modifications of genes. Tumor cell growth requires energy (the Warburg effect) to support cytomatrix mechanics and biosynthesis (Figure 8B).

These physical and chemical processes are accompanied by observable morphological changes, including variations in cell size (mass) and shape, adaptation in energy sources, involving aerobic glycolysis, enhanced cytomatrix mechanics characterized by increased cytoplasmic fluctuations, and increased biosynthesis evidenced by nucleolar hypertrophy. The cancer cycle, starting with benign tumors, includes the accumulation of mutations resulting in heterogeneity and a precancerous state that progresses to neoplasm formation. If any of the four necessary conditions for tumor development are absent, tumor progression and growth may slow down.

At the initial precancerous state, the cell number may increase due to elevated proliferation triggered by a single mutation, inflammation, and hormonal signaling, a condition referred to as hyperplasia. The hyperplasia is typically reversible and non-malignant. Next state, dysplasia, involves multiple mutations, abnormal cell morphology, and disrupted differentiation with the tumor microenvironment formation [59], which is a step closer to tumor initiation (Graphical Abstract). In the tumor progression, the presence of extracellular factors, such as the tumor microenvironment and inflammation, is crucial.

### 1.12. Microbial Pathogens in Actin Microfilament Remodeling

Microbial pathogens have developed various strategies to manipulate host cells and facilitate successful infection. For example, some bacteria, such as *Shigella flexneri*, *Listeria monocytogenes*, and various *Rickettsia* species, use actin comet tails to move and spread between host cells. During infection with *Shigella flexneri*, the elongation of actin comet tails is linked to Src and Abl family kinases that depend on N-WASP-mediated globular actin polymerization. Abl kinases phosphorylate N-WASP, which leads to the elongation of actin fibers and contributes to the intracellular movement and intercellular spread of *Shigella* [60].

Infection with *Helicobacter pylori* is the major cause of chronic gastritis, peptic ulcers, and gastric cancer [61]. Highly virulent strains encode a type IV secretion system, which translocates the CagA effector protein into gastric epithelial cells. Using the *H. pylori* pathogen as a model system, it has been demonstrated that Src and Abl kinases collaborate to trigger global rearrangements of actin microfilaments in which cortactin is a major target [62]. Cortactin is involved in regulating the actin microfilament organization [63] such that CagA disrupts gastric epithelial cell polarity by hijacking cortactin [64]. Again, protooncogenic Src and Abl tyrosine kinase families play a central role in various cancers and in the pathogenesis of *H. pylori* [65,66].

*Shigella flexneri* and *H. pylori* infections illustrate how pathogens may utilize proto-oncogenic proteins as both pathogens take advantage of Src and Abl tyrosine kinases. *Shigella flexneri* builds a comet tail from cytosolic G-actin for movement, whereas *H. pylori* promotes global rearrangements of actin microfilaments of the cytomatrix. *H. pylori* infection triggers both acute and chronic inflammation that work in conjunction with proto-oncoproteins. Thus, proto-oncoproteins can be activated by genetic and epigenetic aberrations as well as inflammatory and pathogenic influences, resulting in cytomatrix modification.

Thus, the deregulation of actin microfilament integration into the solid phase disrupts the architecture of the cytomatrix, resulting in changes in cell shape and size, as well as the process of dedifferentiation. The differences in the effects of *S. flexneri* and *H. pylori* on actin remodeling indicate that the incorporation of actin microfilaments into the solid phase of the cytomatrix is essential for cell differentiation and for maintaining the proper shape and size of cells. This supports the idea that changes to actin microfilaments alone are not enough to cause morphological alterations in cells.

Disruption of the solid phase cytomatrix architecture that results in dedifferentiation is essential for the development of a tumor. This helps explain why clinical cancer is relatively rare, despite the frequent occurrence of gene mutations. Understanding the physical and chemical aspects of cellular chemistry is crucial for identifying the fundamental mechanisms that drive cancer, thereby paving the way for innovative treatments and strategies to combat the disease.

Actual tumor formation is a complex process that proceeds through stages depending on the mutational sites and burden, as well as the status of the cytomatrix. Tumors may contain actively or slowly dividing cells, depending on size and mass accumulation, which also includes quiescent, senescent, and dead (apoptotic and necrotic) cells. The physical morphology of a cancer cell depends on the combination of microfilament activity, cytomatrix mechanics, energy adaptation, and the rate of chemical reactions (metabolism), with results from the presence of more than 800 potential cancer-driving genes that have been identified to date [67].

A wide range of genes and gene mutations are involved in malignant transformation [68]. Malignant success is related to the ability of the cytomatrix to increase the cell’s energy to undergo and survive with the structural changes associated with various stages of differentiation (from benign to different cancerous stages), influenced by the tumor microenvironment.

### 1.13. Role of Inflammation in Tumor Progression

Gastric cancer is a neoplasm linked to the infectious agent *H. pylori*, which is considered to be a primary cause, although microbial, environmental, and host factors modulate the pathogen’s effects [61,69]. *H. pylori* elicits both acute and chronic inflammation as well as intrinsic mediators of inflammatory responses, including proinflammatory cytokines and reactive oxygen and nitrogen species, which can induce both genetic and epigenetic alterations [70].

The role of pro-inflammatory cytokines of acute inflammatory (IL-6, IL-8, TNF-a), and chronic inflammatory process (IL-1, IFN-g) in carcinogenesis have been well characterized [71,72,73,74]. Levels of expression of IL-6, IL-7, IL-8, IL-10, and TNF-α mRNA are significantly higher in mucosa infected with *H. pylori* than in *H. pylori*-negative patients [75]. IL-8 receptors CXCR1 and CXCR2 belong to the G protein-coupled receptor (GPCR) family and play crucial roles in inflammation, angiogenesis, and tumor growth [76]. Once IL-8 binds to its receptor, small GTPases Ras/Rac/Rho/cdc42/Rap1, PKC, and AKT, that exist downstream of the receptor, regulate actin polymerization [77]. On the other hand, the cytokine IL-6 tunes the actin cytoskeleton via STAT3-mediated signaling [78].

Actin microfilament dynamics participate in tissue homeostasis and act as sensors, leading to an immune-mediated anti-cancer response where IFNs drive multiple mechanisms that promote inflammatory signals, but may also initiate feedback suppression of immune cells [79]. In response to TNFα, actin networks modulate NF-κB dynamics, thereby controlling inflammation [80]. IL-1a modulates actin network and cell junctions through the Rho GTPase family [81]. Thus, the clinical outcome of *H. pylori* infection is determined by a complex interplay of host–pathogen interactions.

Taken together, pro-inflammatory factors, oncoproteins, microbial pathogens, and tumor microenvironment alter normal actin dynamics, resulting in morphological remodeling of the cytomatrix. This remodeling results in cell polarity disruption and alterations in cell size, which affect the rate of metabolic reactions and may contribute to malignant transformation and tumor progression in conjunction with the tumor microenvironment.

### 1.14. Role of the Cytomatrix in Tumor Microenvironment Formation

Tumor cells do not exist in isolation in vivo, and tumor development from malignant cells depends on the surrounding tumor microenvironment (TME) composed of various cell types and biophysical and biochemical components and which may have a bimodal effect in that they may initially prevent tumor progression, but as the disease progresses, they could promote tumor growth [82]. The TME typically comprises immune cells, including T and B lymphocytes [83], tumor-associated macrophages [84], dendritic cells, natural killers [85], neutrophils, and myeloid-derived suppressor cells; stromal cells, such as cancer-associated fibroblasts [86], stromal myofibroblasts [87], pericytes, and mesenchymal stromal cells, the extracellular matrix (ECM), and the blood and lymphatic vascular networks, which are collectively enmeshed and in communication with each other and with the heterogeneous cancer cells [88,89].

A series of cytokines, chemokines, growth factors, exosomes, and signaling molecules produced by immune and stromal cells interact with each other and constitute a network within the TME to give the tumor the ability to sustain and survive [90]. A pro-inflammatory microenvironment favors the development of a desmoplastic tumor microenvironment that is highly fibrotic [91]. Desmoplastic reaction (DR), promoted by stromal fibrosis, plays a crucial role in cancer progression [92]. DR refers to the growth of excessive stromal fibrous or connective tissue surrounding tumors, where cancer-associated fibroblasts (CAFs) play a key role in the remodeling of the ECM. The key components of the immunosuppressive TME in solid tumors include CAFs and associated DR tissues. CAFs have a role in creating the extracellular matrix (ECM) and immune reprogramming of the tumor microenvironment [93].

Cancer cells may transform normal fibroblasts (NFs) into CAFs through exosomes. Exosomes from cancer cells, carrying miRNAs and lncRNAs, contribute to the transformation of NFs into CAFs, mediated by pathways including TGF-b/Smads, JAK/STAT, and MAP [94]. Key target genes involved in the transition (from NFs to CAFs) include VEGF for angiogenesis, GLUT1 for glucose metabolism, Lysyl Oxidase for extracellular matrix remodeling, and Carbonic Anhydrase 9 for pH regulation [95].

Tumor microenvironment formation is a dynamic process where cancer cells recruit normal host cells to create a tumor-associated niche. However, the mechanisms and reasons why cancer cells begin recruiting host cells and trigger the generation of the extracellular matrix (ECM) are not well understood.

Revealing a two-phase system of cellular structure and differentiated protein synthesis within these compartments suggests a mechanism for the development of immune responses and inflammatory reactions, as it links these processes to intracellular triggers that initiate desmoplastic reactions. RNA-seq, Ribo-seq, and mass spectrometry analyses of HCT-15 [4,5] demonstrated that the intracellular cytomatrix is integrated with the extracellular matrix (Figure 9).

The KEGG enrichment plots of the cytomatrix illustrate the connection between the intracellular matrix and the ECM. In addition, post-translational modifications (PTMs), such as N- and O-glycan glycosylation, including the metabolism of glycosaminoglycans, glycosphingolipids, glycolipids, and glycerophospholipids, and others, take place in the cytomatrix. These are essential for the formation of cell surface proteoglycans and glycolipoproteins, which mature in the cytomatrix (Figure 10) before being translocated to the cell surface as part of the ECM, glycocalyx, and various cell surface antigens and receptors.

Under normal physiological conditions, the ECM proteoglycans, glycoproteins, and glyco-lipo-proteins are markers of the tissue cells and play crucial roles in cell-to-cell recognition, cell signaling, and immune surveillance and response.

In cancers, multiple mutations might trigger not only the conformational changes of ECM proteins but also the post-translational modification spectrum of glycosylated proteins due to amino acid replacements. In addition, expulsion of intracellular cytomatrix proteins to the cell surface might also contribute to the immune response. The connection between intracellular and extracellular matrices provides an opportunity to predict the antigenic makeup of cancer cells. Predicting changes in the cell surface antigens through analyzing modifications in the cytomatrix proteome could help identify biomarkers for early diagnosis. Studying the timing of antigen exposure on the cell surface might reveal the earliest signs of cancer biomarkers. Additionally, the predicted antigenic profile of cancer cells could assist in developing personalized immunotherapy.

Translocation of the mutated ECM proteins and intracellular structures to the cell surface and localization within the ECM could trigger an immune response, attract diverse TME cells, and activate CAFs, which generate desmoplastic reactions forming dense, fibrous connective tissue around the tumor. DR by insulating malignant cells forms a closed system that disrupts normal signal transduction, dysregulating cytomatrix actin dynamics. Non-muscle actomyosin works as a pump, like the heart, circulating the cytosol, delivering substrates and nutrients to the immobilized catalytic complexes. Like a heartbeat, intracellular actin dynamics should have a rhythm; the excess of it could increase the rate of cellular chemistry.

Insulation by TME fibers, interrupting the normal rhythm of chemical processes, facilitates a high rate of metabolism in cancerous cells, probably, by the self-polymerization properties of globular actin in the presence of ATP, providing self-sustained cytomatrix dynamics. This extreme metabolism results in typical cancer cell morphology. In addition, a closed system with a limited number of cells surrounded by a desmoplastic microenvironment (fibers and fat) may have a good chance of surviving in the bloodstream and allow spreading through the blood to form metastases. Thus, linking intracellular abnormalities with extracellular changes clarifies how tumors progress from malignant cells, in which TEM and inflammatory factors, including pathogens, play essential roles in tumor formation and metastasis.

## 2. Perspective

Historically, biological understanding has been advanced through a reductionist approach, where scientists studied individual components of a cell, such as a single gene, protein, or metabolic pathway, in isolation. While this method led to groundbreaking discoveries, it presents a fragmented view of cellular life. The concept of elucidating the cell as a whole, rather than through separate fields such as molecular biology or biochemistry, is challenged by the complexity of the cytoplasm. Revealing that elastic solid (cytomatrix) and viscous liquid (cytosol) phases exist opens the perspective of integrated cell modeling. This holistic approach recognizes that a cell’s behavior is more than the sum of its isolated parts and focuses on how molecular components interact dynamically within the complete system.

The cytomatrix operates in a state that is far from equilibrium and similar to the intricate workings of a microchip; it utilizes energy to maintain its complex structure and essential functions, enabling the cytomatrix to adapt and respond to various biological demands. This type of operation occurs in systems that are not in thermodynamic equilibrium. This phenomenon is referred to as non-equilibrium dynamics. In this context, the behavior of the solid-phase cytomatrix can be studied as a physical system.

The cytomatrix is a dynamic system that possesses the remarkable ability to seamlessly integrate a diverse array of signals, including mechanical, chemical, and spatial cues. In doing so, the cytomatrix effectively performs the sophisticated biological computations that contribute to the regulation and coordination of cellular processes. Comparison of artificial intelligence (AI) with the cytomatrix of living cells provides a useful analogy for understanding the complex mechanisms of both biological and technological information processing.

The biochemical reactions occurring within the cytomatrix are influenced not only by the natural tendency of molecules to diffuse but also by mechanical gating and careful coordination of spatial arrangements. This interplay ensures that reactions remain organized precisely, maximizing the efficiency and effectiveness of cellular operations. Solid-state biochemistry of the cytomatrix could focus on the study of a complex yet static structure, whose architecture involves every intricate detail, including the relationships between individual atoms and molecules and how these arrangements affect the overall strength and stability of the cytomatrix. Exploiting the similarities between the cytomatrix, a complex network that provides structural support and organization within cells, and microchips, which are the core components of AI and neural network machine learning, can significantly enhance our understanding of cellular architecture. This investigation has the potential to transform current knowledge and lead to groundbreaking advancements in cellular biology (Table 1).

Engineering an artificial cytomatrix based on cellular cytomatrix should result in programmable cellular systems as well as elucidate how tumor cells rewire their cytomatrix to alter mechanical signaling and evade regulation.

Current high-throughput methods, such as mass spectrometry and gene set enrichment analysis, can create metabolic pathways from grouped gene sets. These methods effectively demonstrate the differences between the cytosol and cytomatrix. However, utilizing AI modeling in conjunction with neural network-based machine learning should enhance these studies, allowing for the functional validation, interpretation, and prediction of cellular outcomes, particularly in the transformation of healthy cells into malignant cells (Table 2).

By utilizing a two-phase system of cytoplasmic organization, we can enhance the dynamics of mass spectrometry and GSEA datasets when integrated with artificial intelligence (AI). AI approaches should be able to leverage existing knowledge from the literature to harmonize fragmented datasets, allowing for a more apparent distinction between the cytosol and the cytomatrix. This process facilitates a more thorough mechanistic and functional interpretation of high-throughput experiments.

While a simple listing of proteins or pathways can differentiate between the cytosol and cytomatrix, this method offers limited insight into their functional roles and cannot compete with the depth of analysis potentially provided by AI. AI analysis of intracellular and extracellular matrices should provide detailed insights into the mechanisms of tumor microenvironment formation, depending on the mutational burden, as these solid phases are interconnected. To truly understand the complexities of the cytomatrix and cytosol, we must harness advancements in artificial intelligence and machine learning to model and predict cellular processes and cell fate, particularly in normal and pathological developments, such as cancer progression and the effects of the tumor microenvironment (TME) on the formation of cancerous tumors.

For mechanical analysis, the cytomatrix refers to the solid-phase, structural framework within cells, composed of cytoskeletal elements, organelle scaffolds, and molecular complexes that govern spatial organization, mechanical integrity, and biochemical gating. It is not just passive scaffolding; it actively regulates cells’ sense and response to mechanical forces (Mechanotransduction), spatial control of enzymatic reactions and substrates (Gated Metabolic Coupling), and dynamic reconfiguration in response to stress or signaling (Adaptive Remodeling).

For chemical analysis, cytomatrix is an intricate network within the cytoplasm of cells, consisting of structural proteins, ribosomes, and enzymes. The cytomatrix plays a role in various cellular processes, including: a) Organelle formation (Immobilization and segregation of catalytic complexes and proteome can potentially lead to the formation of organelles), b) Immobilized biocatalysis (the cytomatrix contains structural proteins, ribosomes, and enzymes that facilitate unique biosynthetic pathways and immobilized biocatalysis), c) cellular micromechanics and cytoplasmic motion (the cytomatrix contributes to the mechanical dynamics of cells for cytoplasmic movement).

Thus, the cytomatrix refers to the dense, structured mechano-chemical network that behaves more like a solid-phase computational substrate with Microchip-Like Features that can be adapted for neural network modeling.

Employing non-equilibrium thermodynamics methods alongside neural network-based machine learning to analyze the solid-phase cytomatrix and the extracellular microenvironment holds the potential to create a new interdisciplinary field that merges life sciences with AI technologies.

## 3. Conclusions

Non-muscle actin has historically been regarded as a primary component of the cytoskeleton, responsible for maintaining cell shape, regulating contractility, and facilitating cell motility. A careful examination of the actin network, alongside other filaments and associated proteins, shows that microfilaments also orchestrate solid-phase cytomatrix dynamics, which in turn regulate the physicochemical processes of both healthy and cancer cells. Chemical processes within the intracellular environment, a viscous, dense, and heterogeneous structure that restricts free diffusion, are regulated by actin dynamics. Although chemical reactions within cells often differ from those in experimental settings, both are governed by the laws of physics and chemistry. Actin filaments receive numerous regulatory signals from the extracellular environment and transduce these signals to chemical processes by utilizing ATP energy generated by mitochondria. Deregulation of orchestrated signals in malignant cells, due to genetic alterations, the tumor microenvironment, or pathogens, may result in actin microfilament remodeling, which also requires an additional energy source, such as aerobic glycolysis. Actin, as a component of the solid phase cytomatrix, is responsible for altering cell shape and size in malignant cells, which is known as dedifferentiation or morphological alteration. The cytomatrix is a complex structural framework within cells that plays a pivotal role in maintaining cellular integrity and facilitating various biochemical processes. The dynamics of this cytomatrix are significantly influenced by actin regulators, which are proteins that modulate the assembly and disassembly of actin filaments. These regulators ensure that the actin responds effectively to various intracellular and extracellular signals, thereby fine-tuning the chemical reactions that occur within the cell. Moreover, by regulating the organization and movement of the cytomatrix, actin regulators also impact gene expression patterns. This intricate interplay between the cytomatrix structure, actin regulators, and TME presents a valuable opportunity for researchers to explore and gain a deeper understanding of cellular chemistry, offering insights into how cells function, adapt, and respond to their environment. The structure of the cytomatrix and the regulation of actin dynamics within it also present significant challenges and exciting opportunities for future research, particularly regarding their effects on chemical reactions and gene expression under normal physiological conditions and in cancers, as well as in therapeutic contexts. The high metabolic rate of malignant cells significantly impacts gene expression, resulting in distinct differences between cancerous and normal tissues. Cancer-related pathways are prominently represented in the cytomatrix, enabling the clear identification of specific pathways that are abundant in cancers but underrepresented or absent in healthy cells. By targeting these overrepresented proteins and cancer pathways in the cytomatrix that are absent in normal tissue, it is possible to develop highly effective and non-toxic anticancer drugs or combinations that have the potential to transform cancer treatment.

## Figures and Tables

**Figure 1 cancers-17-03686-f001:**
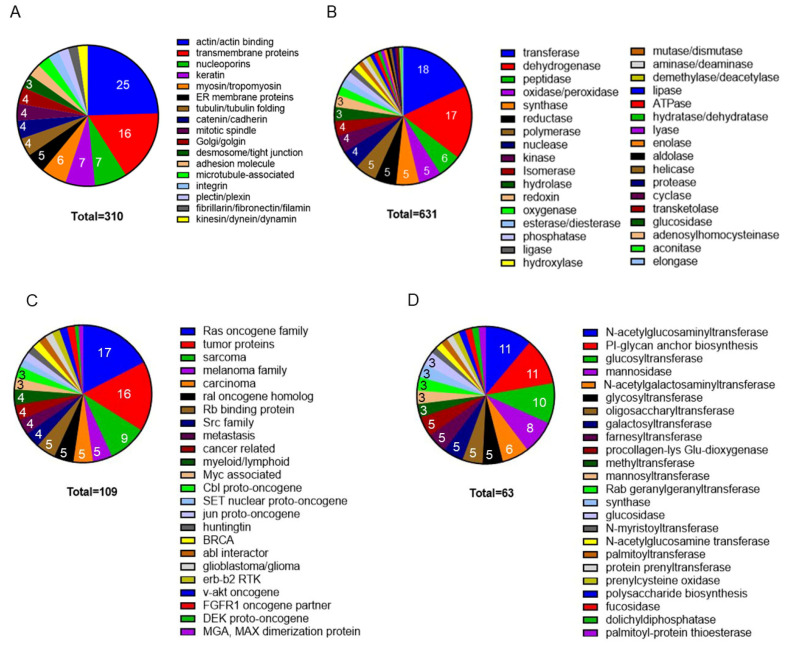
Mass-spectrometry analyses of the cytomatrix proteome. The Pie charts illustrate (**A**) the structural proteome, (**B**) the enzyme spectrum of the cytomatrix of HCT−15 cells, (**C**) Cancer pathways, (**D**) Protein post-translational modification pathways. The proteins in the group were identified according to Intensity-Based Absolute Quantification (iBAQ) scores. The percentage for each group was calculated based on the total number of selected proteomes in each collection, which is shown within the slices. Figures reproduced from T.E. Shaiken et al., iScience (2023) and Shaiken, BBR (2025) [4,5].

**Figure 2 cancers-17-03686-f002:**
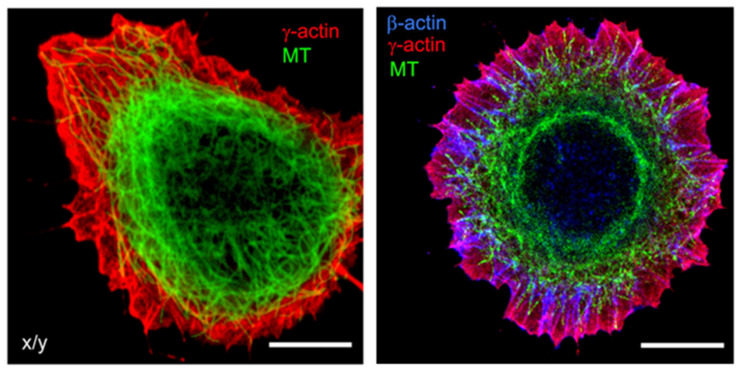
Microfilaments and Microtubules. Subcellular localization of cytoplasmic actins and microtubules in spreading epithelial HaCaT cells. Cells were stained for β−actin (blue), γ−actin (red), and α−tubulin (green). Bars, 10 μm. Figures reproduced from V. Dugina et al., Oncotarget (2016) [9].

**Figure 3 cancers-17-03686-f003:**
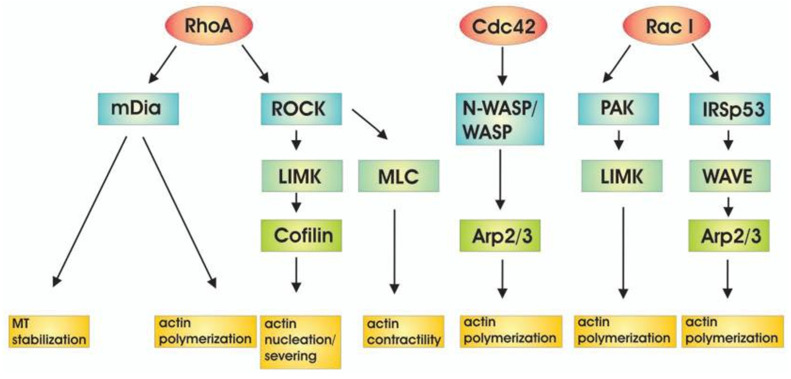
Downstream targets of Rho family of GTPases lead to actin regulation. Figure reproduced from D. Spiering and L. Hodgson, Cell Adhesion & Migration (2011) [14].

**Figure 4 cancers-17-03686-f004:**
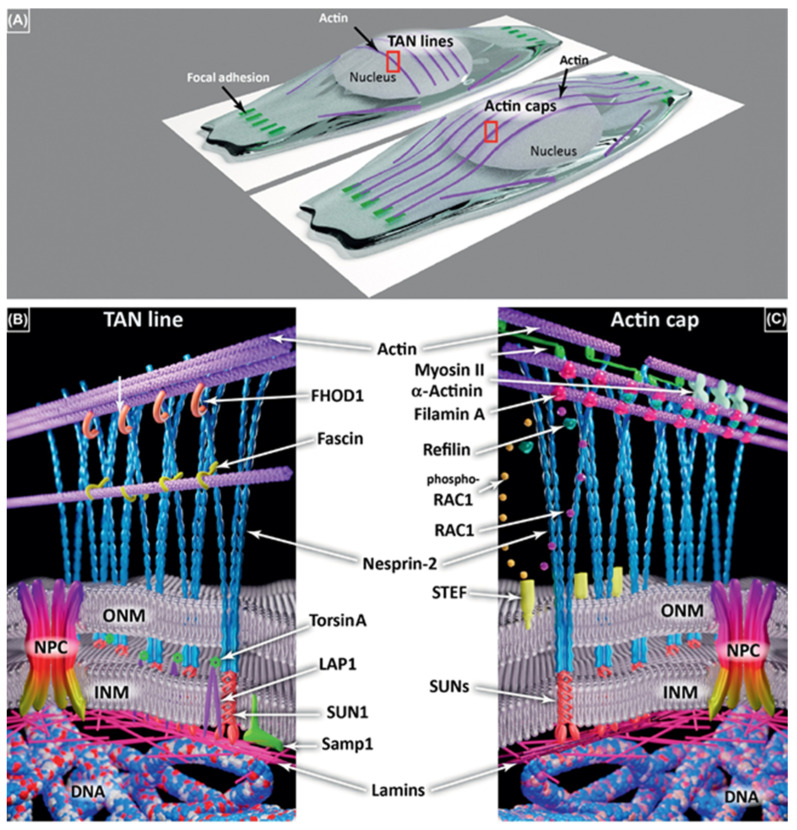
Transmembrane Actin−Associated Nuclear (TAN) Lines and Perinuclear Actin Caps. (**A**) TAN lines are associated with actin transverse arcs, parallel with the edge of the cell front, and not connected to focal adhesions. Actin caps are longitudinal actin fibers initiated at focal adhesions. Both require the linker of the nucleoskeleton and the cytoskeleton (LINC) complex comprising Nesprins (blue) and Sad1p, UNC-84 (SUN) proteins (red) in (**B**,**C**). (**B**) Several partners of the TAN lines have been identified on both sides of the nuclear envelope (NE) including formin homology 2 domain-containing 1 (FHOD1) (orange), fascin (yellow), TorsinA (light green), lamin-associated polypeptide 1 (LAP1) (purple), and Samp1 (dark green). (**C**) Various partners are involved in actin cap formation and maintenance, including myosin II (fluo−green), α−actinin (light green), and Refilin (dark green) that interacts with filamin A (red) in a Rac1 (blue)/phospho-Rac1 (dark yellow)—dependent manner driven by Sif and Tiam1-like exchange factor (STEF) (light yellow). Both actin caps and TAN lines are connected to lamins (pink) and then DNA (blue-white-red ribbon). Figures reproduced from P.M. Davidson and B. Cadot, Trends in Cell Biology (2021) [10].

**Figure 5 cancers-17-03686-f005:**
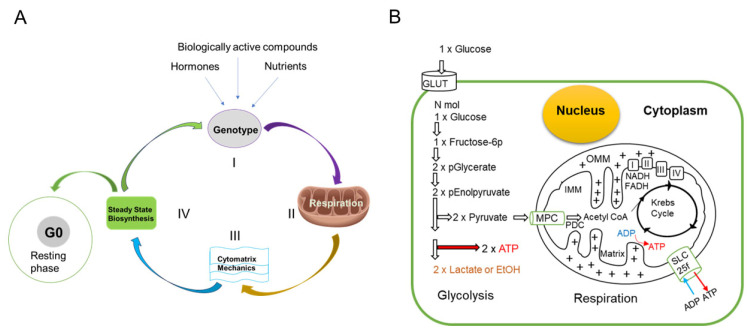
Life Cycle of Healthy Cells and Glucose Metabolism. (**A**) In healthy cells, depending on the cell type, external signaling triggers gene expression (Step I). Oxidative phosphorylation consistently produces ATP (Step II) for normal cellular functions, including the mechanics of the cytomatrix (Step III) that results in cytoplasmic fluctuations that help sustain steady-state biosynthetic processes and metabolism (Step IV). (**B**) Schematic representation of the relationship between glycolysis and oxidative phosphorylation. Glucose uptake and ATP generation pathways. Insulin and glucose transporters regulate glucose uptake; MPC—mitochondrial pyruvate carrier; PDC—pyruvate dehydrogenase complex, IMM—inner and OMM—outer mitochondrial membrane, I, II, III, IV—electron transport chain, SLCf—solute carrier family, GLUT—glucose transporter. The arrows indicate the direction of the process. A mitochondria diagram with protons (+) shows protons being pumped from the mitochondrial matrix to the intermembrane space, creating a proton gradient. This proton force is used by ATP synthase to drive the synthesis of ATP as protons flow back into the matrix.

**Figure 6 cancers-17-03686-f006:**
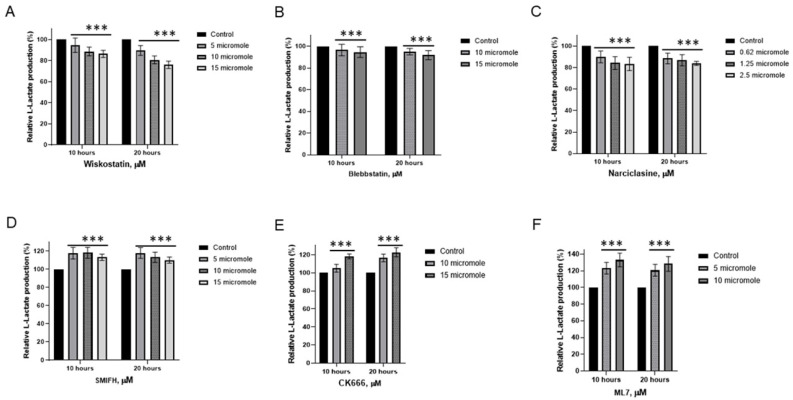
Effects of actin modulators on L-Lactate generation in HCT−15 colon cancer cells. (**A**) Wiskostatin, (**B**) Blebbistatin, (**C**) Narciclasine, (**D**) SMIFH, (**E**) CK666, (**F**) ML7. The data represent the mean ± SD of four replicates. *** *p* < 0.001 compared to untreated. Figure reproduced from T.E. Shaiken, BBR (2025) [5].

**Figure 7 cancers-17-03686-f007:**
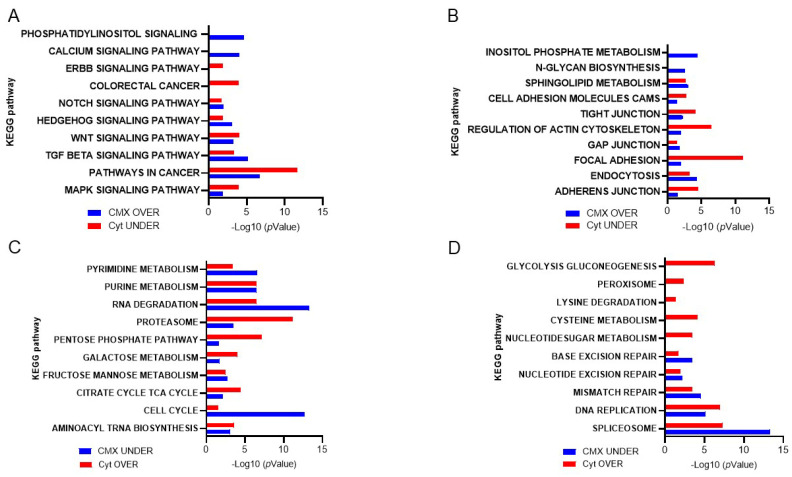
KEGG−based analysis of ribosome footprints of colorectal cancer HCT−15 cells. Overrepresented pathways in the cytomatrix and underrepresented pathways in the cytosol are shown. (**A**) Intracellular pathways that regulate actin dynamics. (**B**) Extracellular signaling that regulates actin dynamics. (**C**,**D**) Overrepresented cytosolic and underrepresented cytomatrix pathways. Figures reproduced from T.E. Shaiken, BBR (2025) [5].

**Figure 8 cancers-17-03686-f008:**
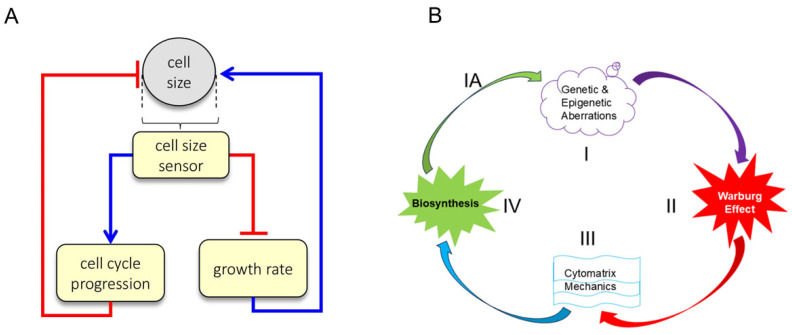
Dual-mechanism model of cell size specification and Cancer Cycle. (**A**) Cells employ two separate strategies to correct deviations from their appropriate size: small cells spend more time in G1 and small cells grow faster than large cells. Figure 5A is reproduced from M.B. Ginzberg et al., eLife (2018) [52]. (**B**) Malignant transformation of healthy cells to cancerous cells. The cancer cycle starts with genetic alterations (Step I) that alter cell homeostasis, which may lead to energy-producing aerobic glycolysis (Step II), consequently elevated cytomatrix mechanics (Step III) and accelerated biochemical processes (Step IV), resulting in biomass accumulation and cell size (mass) increases, which trigger cell division or a senescent state. Progression and accumulation of mutations can lead to heterogeneity and the development of a benign or malignant tumor (Step IA). The absence of any step in the cancer cycle may suppress the development of clinical tumors. Figure 5B reproduced from T.E. Shaiken, BBR (2025) [5].

**Figure 9 cancers-17-03686-f009:**
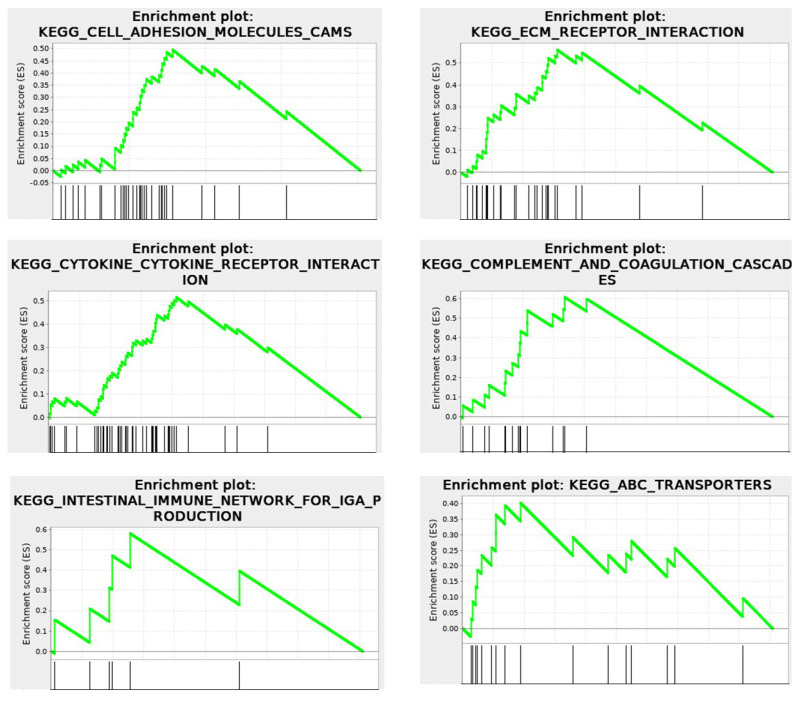
RNA−seq. KEGG Enrichment plots for Cell Surface events, illustrating connections of intracellular and extracellular matrices. Cytomatrix vs. Cytosol. 

 Enrichment profile; 

 Hits.

**Figure 10 cancers-17-03686-f010:**
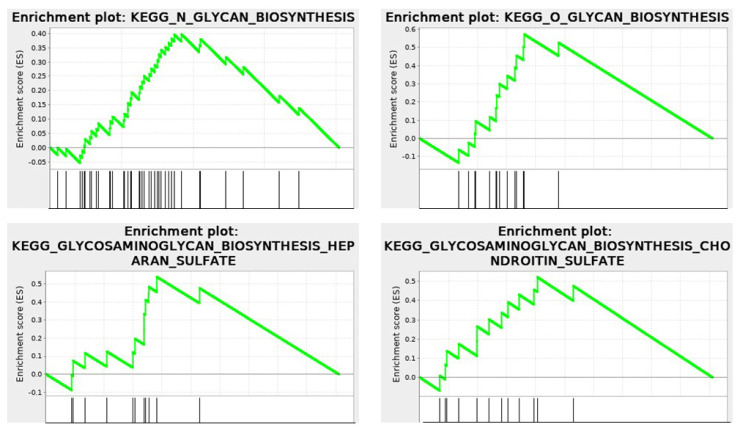
RNA−seq. KEGG Enrichment plots show that metabolism and glycosylation PTMs occur within the cytomatrix. Cytomatrix vs. Cytosol.

**Table 1 cancers-17-03686-t001:** The parallel between Microchips and the Cytomatrix.

Feature	Cytomatrix Function	Microchip Analogy
Spatial Encoding	Organizes enzymes, ribosomes, and signaling hubs	Circuit layout and logic gates
Mechanical Gating	Responds to tension, compression, and shear forces	Transistor-like switching via mechanical input
Signal Propagation	Directs biochemical signals along cytoskeletal tracks (actins)	Electrical signal routing in wires
Compartmentalization	Creates microdomains (organelles) for localized reactions	Memory blocks and processor cores
Energy Coupling	Links ATP production to mechanical strain	Power distribution across chip architecture

**Table 2 cancers-17-03686-t002:** AI neural network-based machine learning models for cytomatrix study.

Process	Description
EMT Switch	Simulate epithelial-to-mesenchymal transition with cytomatrix reprogramming
Drug Perturbation	Model effects of cytoskeletal drugs (e.g., paclitaxel, latrunculin)
Cancer Cell Module	Simulate cytomatrix softening, polarity loss, and invasive mechanics
Machine Learning	Predict cytomatrix configurations that optimize signaling or resist deformation

## Data Availability

The RNA sequencing data were uploaded to NCBI GEO (GSE199535), and mass spectrometry proteomics data have been deposited to the ProteomeXchange Consortium via the PRIDE partner repository (PXD031043).

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
