# Peer review of "Elastic Cytomatrix Dynamics Influences Metabolic Rate and Tumor Microenvironment Formation"

_cancers, 2025, doi:10.3390/cancers17223686_

Round 1
Reviewer 1 Report
Comments and Suggestions for Authors
I carefully read and reviewed the paper titled “Elastic Cytomatrix Dynamics Influences Metabolic Rate and Tumor Microenvironment Formation”. Authors overviewed explores the complex interplay between cytomatrix mechanics, mitochondrial respiration, glycolytic metabolism, and tumor microenvironment (TME) formation. They proposed an integrated biophysical and biochemical perspective where the intracellular cytomatrix and extracellular matrix (ECM) function as cooperative elastic solid phases influencing cancer development and progression. The conceptual linkage between altered protein dynamics, actin remodeling, immune cell attraction, and niche formation offered an innovative interpretation of tumor insulation and dedifferentiation. Additionally, the discussion on AI applications in cytomatrix modeling brings a modern and forward-looking angle to cancer biology research.
However, several issues must be addressed:
* Authors outlined pathophysiological mechanisms but they didn’t adequately translate them into potential intervention strategies targeting cytomatrix or ECM mechanics. Discuss please.
* The section on neural network modeling appears separate from the main biological narrative. Elaborating concrete AI applications including mechanical modeling, spatial transcriptomics integration, would strengthen relevance. Discuss please.
* Tumor stroma activation is recognized but positioning ECM insulation as a primary cause of immune evasion and dedifferentiation may be speculative without empirical grounding. Elaborate on please.
Author Response
Reviewer Comments and Suggestions for Authors
I carefully read and reviewed the paper titled “Elastic Cytomatrix Dynamics Influences Metabolic Rate and Tumor Microenvironment Formation”. Authors overviewed explores the complex interplay between cytomatrix mechanics, mitochondrial respiration, glycolytic metabolism, and tumor microenvironment (TME) formation. They proposed an integrated biophysical and biochemical perspective where the intracellular cytomatrix and extracellular matrix (ECM) function as cooperative elastic solid phases influencing cancer development and progression. The conceptual linkage between altered protein dynamics, actin remodeling, immune cell attraction, and niche formation offered an innovative interpretation of tumor insulation and dedifferentiation. Additionally, the discussion on AI applications in cytomatrix modeling brings a modern and forward-looking angle to cancer biology research.
Response: Thank you very much for taking the time to review the manuscript. We thank the Reviewer for the effort, comments, and suggestions, which helped us improve the content and presentation of the manuscript. Please find the detailed responses below. The corresponding revisions are highlighted in the resubmitted manuscript file.
However, several issues must be addressed:
Comments 1
* Authors outlined pathophysiological mechanisms but they didn’t adequately translate them into potential intervention strategies targeting cytomatrix or ECM mechanics. Discuss please.
Response 1: Thanks for pointing out the importance of the intervention strategy.
In the section Conclusions, lines 706–715 (page 19 and 20), we discuss this as follows: The high metabolic rate of malignant cells significantly impacts gene expression, resulting in distinct differences between cancerous and normal tissues. Cancer-related pathways are prominently represented in the cytomatrix, enabling the clear identification of specific pathways that are abundant in cancers but underrepresented or absent in healthy cells. By targeting these overrepresented proteins and cancer pathways in the cytomatrix that are absent in normal tissue, it is possible to develop highly effective and non-toxic anticancer drugs or combinations that have the potential to transform cancer treatment.
Lines 650–656 (pages 18) were added to emphasize the practical aspect of the link between the cytomatrix and ECM: The connection between intracellular and extracellular matrices provides an opportunity to predict the antigenic makeup of cancer cells. Predicting changes in the cell surface antigens through analyzing modifications in the cytomatrix proteome could help identify biomarkers for early diagnosis. Studying the timing of antigen exposure on the cell surface might reveal the earliest signs of cancer biomarkers. Additionally, the predicted antigenic profile of cancer cells could assist in developing personalized immunotherapy.
Comments 2
* The section on neural network modeling appears separate from the main biological narrative. Elaborating concrete AI applications including mechanical modeling, spatial transcriptomics integration, would strengthen relevance. Discuss please.
Response 2: Agree, we have extended the section 'Perspectives' to explain how the cytomatrix can be adapted for AI/neural network machine learning.
In lines 781 – 795 (pages 21 and 22), we discuss it as follows: For mechanical analysis, the cytomatrix refers to the solid-phase, structural framework within cells, composed of cytoskeletal elements, organelle scaffolds, and molecular complexes that govern spatial organization, mechanical integrity, and biochemical gating. It’s not just passive scaffolding; it actively regulates cells’ sense and response to mechanical forces (Mechanotransduction), spatial control of enzymatic reactions and substrates (Gated Metabolic Coupling), and dynamic reconfiguration in response to stress or signaling (Adaptive Remodeling).
For chemical analysis, cytomatrix is an intricate network within the cytoplasm of cells, consisting of structural proteins, ribosomes, and enzymes. The cytomatrix plays a role in various cellular processes, including: a) Organelle formation (Immobilization and segregation of catalytic complexes and proteome can potentially lead to the formation of organelles), b) Immobilized biocatalysis (the cytomatrix contains structural proteins, ribosomes, and enzymes that facilitate unique biosynthetic pathways and immobilized biocatalysis), c) cellular micromechanics and cytoplasmic motion (the cytomatrix contributes to the mechanical dynamics of cells for cytoplasmic movement).
Thus, the cytomatrix refers to the dense, structured mechano-chemical network that behaves more like a solid-phase computational substrate with Microchip-Like Features that can be adapted for neural network modeling.
In addition, at lines 747– 752(page 20), we improved the text as follows: Exploiting the similarities between the cytomatrix, a complex network that provides structural support and organization within cells, and microchips, which are the core components of AI and neural network machine learning, can significantly enhance our understanding of cellular architecture. This investigation has the potential to transform current knowledge and lead to groundbreaking advancements in cellular biology.
Comments 3
* Tumor stroma activation is recognized but positioning ECM insulation as a primary cause of immune evasion and dedifferentiation may be speculative without empirical grounding. Elaborate on please.
Response 3: In the manuscript (lines 615 – 621, page 17), we cited several works related to immune suppression by CAFs as shown below:
Desmoplastic reaction (DR), promoted by stromal fibrosis, plays a crucial role in cancer progression [92]. DR refers to the growth of excessive stromal fibrous or connective tissue surrounding tumors, where cancer-associated fibroblasts (CAFs) play a key role in the remodeling of the ECM. The key components of the immunosuppressive TME in solid tumors include CAFs and associated DR tissues. CAFs have a role in creating the extracellular matrix (ECM) and immune reprogramming of the tumor microenvironment [93]. Immune evasion was mentioned only in Table 2. To avoid confusion, the Immune Evasion row was removed from Table 2, and Lines 668–670 (page 19) were revised as follows: DR by insulating malignant cells forms a closed system that disrupts normal signal transduction and dysregulates cytomatrix actin dynamics. Original text: DR by insulating malignant cells forms a closed system that disrupts normal signal transduction, dysregulates cytomatrix actin dynamics that may eventually alter the shape and size of tumor cells.
Sincerely,
Tattym E. Shaiken
Reviewer 2 Report
Comments and Suggestions for Authors
This review article elaborates on the roles of elastic cytomatrix on metabolic rate and tumor microenvironment formation. The manuscript is written moderately in logical and practical aspects, but some key aspects are missing. Since, some practical issues need to be addressed before publication. So, I suggest minor revisions.
1. Too many keywords. Revise following the journal guidelines.
2. The starting of the Introduction sounds non-scientific. Please revise them. And there is no need to add subheadings in the Introduction Section.
3. A clear outline of the review in the introduction needs to be added.
4. The explanation for the pathways seems insufficient. The authors can extend this part with suitable figures and descriptions.
5. Only few figures in the draft. Relevant figures related to the link with the target pathways should be added and improved.
6. The information on the tables is insufficient. I suggest extending the table with suitable references.
Author Response
Reviewer Comments and Suggestions for Authors
This review article elaborates on the roles of elastic cytomatrix on metabolic rate and tumor microenvironment formation. The manuscript is written moderately in logical and practical aspects, but some key aspects are missing. Since, some practical issues need to be addressed before publication. So, I suggest minor revisions.
Response: Thank you for reviewing the manuscript. We appreciate the reviewer's time and effort, as well as comments and suggestions, which have helped us enhance both the content and presentation. Below, you will find our detailed responses, and the corresponding revisions are highlighted in the resubmitted manuscript file.
Comments:
- Too many keywords. Revise following the journal guidelines.
Response 1: Seven Keywords: cytomatrix; Warburg effect; actin; tumor microenvironment; metabolism; extracellular matrix; malignant transformation.
The starting of the Introduction sounds non-scientific. Please revise them. And there is no need to add subheadings in the Introduction Section.
Response 2: The start of the introduction (lines 48 – 58, page 2) was revised as follows: Every living organism, from bacteria to humans, has a distinct, recognizable physical shape, and this also applies to every cell in a tissue at the microscopic level, which performs a specific function because of its physical attributes. Disruption of the cell's physical morphology, known as dedifferentiation, is often observed in cancers. The role of the solid phase in maintaining cell morphology as a structural and functional anchor within a large liquid environment remains to be elucidated. It remains uncertain whether cells need a solid phase—defined as a solid component that serves as a stationary or solid-bound phase distinct from the liquid phase—to maintain cell shape and size. The prevailing view is that the cytoskeleton, which consists of cytoplasmic filaments including microfilaments, intermediate filaments, and microtubules, provides structural support and helps maintain cell shape.
A clear outline of the review in the introduction needs to be added.
Response 3: The introduction section was extended (lines 68 – 85, page 2) as follows:
In this review, we examined the biochemical composition of the cytomatrix and its connection to the extracellular matrix (ECM), focusing on the unique features of chemical processes within the cytoplasm and the role of microfilaments in regulating metabolic rates. We also analyzed the energy needs of both healthy and malignant cells in relation to the dynamics of the cytomatrix, explaining the Warburg effect as an adaptation to the rapid fluctuations in energy observed in cancer cells due to solid-phase mechanics. Summarizing known traits of cancer cells and our findings regarding gaps in cell structure (as a two-phase system), along with the role of the Warburg effect in non-muscle actomyosin dynamics, we introduced the Cancer Cycle concept. This concept integrates genetic alterations linked to cancer with the physicochemical processes occurring in the cytoplasm. The intracellular cytomatrix and extracellular matrix, functioning as an integrated solid-phase system, regulate the metabolic rates of both healthy and malignant cells. Changes in the ECM of malignant cells, driven by mutations in the cytomatrix, may attract immune and stromal cells, contributing to tumor microenvironment formation. Finally, we explored the potential of artificial intelligence and neural machine learning in interpreting cells as structured solids containing liquids, which could lead to a new interdisciplinary field combining life sciences with AI technologies.
The explanation for the pathways seems insufficient. The authors can extend this part with suitable figures and descriptions.
5. Only few figures in the draft. Relevant figures related to the link with the target pathways should be added and improved.
Response 4 and 5: New figures have been added, and the following modifications have been made to the existing figures:
- Figure 1 was modified by adding panels C and D, which show cancer and protein posttranslational pathways.
- New Figure 2, which shows the subcellular localization of actins and microtubules.
- New Figure 3, which shows the downstream targets of the Rho family of GTPases related to actin regulation.
- New Figure 4, which shows Transmembrane Actin-Associated Nuclear (TAN) Lines and Perinuclear Actin Caps.
- Figure 7 was modified by adding panels C and D, which show overrepresented pathways of the cytosol.
The manuscript includes the corresponding text explaining the figures. All figures and images are licensed under a CC BY 4.0 license. (http://creativecommons.org)
The information on the tables is insufficient. I suggest extending the table with suitable references.
Response 6: The following text (lines 781 – 795, pages 21 and 22) has been incorporated for tables.
For mechanical analysis, the cytomatrix refers to the solid-phase, structural framework within cells, composed of cytoskeletal elements, organelle scaffolds, and molecular complexes that govern spatial organization, mechanical integrity, and biochemical gating. It’s not just passive scaffolding; it actively regulates cells’ sense and response to mechanical forces (Mechanotransduction), spatial control of enzymatic reactions and substrates (Gated Metabolic Coupling), and dynamic reconfiguration in response to stress or signaling (Adaptive Remodeling).
For chemical analysis, cytomatrix is an intricate network within the cytoplasm of cells, consisting of structural proteins, ribosomes, and enzymes. The cytomatrix plays a role in various cellular processes, including: a) Organelle formation (Immobilization and segregation of catalytic complexes and proteome can potentially lead to the formation of organelles), b) Immobilized biocatalysis (the cytomatrix contains structural proteins, ribosomes, and enzymes that facilitate unique biosynthetic pathways and immobilized biocatalysis), c) cellular micromechanics and cytoplasmic motion (the cytomatrix contributes to the mechanical dynamics of cells for cytoplasmic movement).
Sincerely,
Tattym E. Shaiken